# Conformation-dependent degradation of thermally activated delayed fluorescence materials bearing cycloamino donors

Sanju Hwang[1], Yu Kyung Moon[1], Ho Jin Jang [2], Sinheui Kim[1], Hyein Jeong[3], Jun Yeob Lee [2✉] & Youngmin You [1✉]

Organic light-emitting devices (OLEDs) containing organic molecules that exhibit thermally activated delayed fluorescence (TADF) produce high efficiencies. One challenge to the commercialization of the TADF OLEDs that remains to be addressed is their operational stability. Here we investigate the molecular factors that govern the stability of various archetypal TADF molecules based on a cycloamino donor–acceptor platform. Our results reveal that the intrinsic stability depends sensitively on the identity of the cycloamino donors in the TADF compounds. The rates and photochemical quantum yields of the degradation are positively correlated with the operation lifetimes of the devices. Our research shows that the stability is governed by the conformeric heterogeneity between the pseudo-axial and pseudo-equatorial forms of the cycloamino donor. Spontaneous bond dissociation occurs in the former (i.e., the pseudo-axial form), but the cleavage is disfavored in the pseudo-equatorial form. These findings provide valuable insights into the design of stable TADF molecules.

[1] Division of Chemical Engineering and Materials Science, Ewha Womans University, Seoul 03760, Republic of Korea. [2] School of Chemical Engineering, Sungkyunkwan University, Suwon, Gyeonggi-do 16419, Republic of Korea. [3] Display Research Center, Samsung Display, Yongin, Gyeonggi-do 17113, Republic of Korea. ✉email: leej17@skku.edu; odds2@ewha.ac.kr

Organic light-emitting devices (OLEDs) depend on the utilization of electrogenerated excitons for luminescence. Molecules that display thermally activated delayed fluorescence (TADF) have emerged as promising emitting materials for OLEDs because they enable exciton harvest without relying on precious transition metals[1–4]. High electroluminescence efficiencies resulting from facile exciton harvest have stimulated intense research into the development of OLEDs based on TADF molecules[5–7]. One drawback of TADF OLEDs that retards the exploitation of their full potential is their short operational lifetime. In analogy with phosphorescent devices[8–10], TADF OLEDs exhibit problems associated with the gradual decrease in electroluminescence performance during normal operation[11]. This instability originates from the irreversible degradation of organic materials. The accumulation of degradation byproducts deteriorates device performance because such byproducts quench luminescent excitons, generate non-emissive excitons, and trap charge carriers.

Several recent studies have linked operational stability of the devices with the intrinsic degradation of TADF materials. Adachi and co-workers found the degradation of a TADF dopant and a 3,3-di(9H-carbazol-9-yl)biphenyl (mCBP) host upon the application of electrical and photolytic stresses[12]; it was proposed that excitons are the key intermediates in these degradations. In order to lower exciton densities, the Adachi group incorporated [Li(2-quinolate)] (Liq) exciton-quenching layers in the vicinity of the emitting layer[13]. It has also been shown that intrinsic degradation is further accelerated by bimolecular annihilation involving excitons and polarons[14–17]. These bimolecular processes can be suppressed through the judicious control of an exciton formation zone or by balancing charge carrier transport in the emitting layer[18,19]. Alternatively, decreasing the exciton lifetime of the TADF compound is a viable approach to minimizing bimolecular degradation[20].

The research outlined above coherently points to a strong correlation between device instability and the level of excitons and polarons accumulated within organic layers in devices. Although this correlation has guided the development of novel device strategies, their utility remains limited. This limitation is due to the paucity of our understanding of the structural factors governing the intrinsic stability of materials. Although recent research has investigated the influence of structural modifications on stability[20,21], the molecular parameters that govern intrinsic stability have to date been poorly elucidated. Considering their enormous potential, it is thus of profound importance to establish the structure–stability relationship of TADF molecules. This challenge motivated us to attempt to identify the molecular factors that favor the intrinsic longevity of TADF compounds.

To enable this study, a series of dyads consisting of a 2,4-diphenyl-1,3,5-triazine (TRZ) acceptor (A) and a variety of cycloamino donors (Ds) are chosen. This archetypal D–A framework has been used successfully to create TADF dopant molecules for high-efficiency OLEDs[22–24]. We perform investigations based on structural, photochemical, and quantum chemical techniques, as well as device experiments. Our results reveal that device lifetime is proportional to the intrinsic stability of the TADF dopant. More importantly, we find that the stability of these devices is intimately linked to the equilibrium between the pseudo-axial and pseudo-equatorial conformers of the cycloamino donors. The pseudo-axial conformer allows for exergonic bond dissociation from the singlet intramolecular charge-transfer ($^1$ICT) transition state, whereas this cleavage is thermodynamically disfavored in the pseudo-equatorial form. This conformeric heterogeneity is known to be closely linked to the emergence of TADF[25–31]. However, its effect on the intrinsic stability of TADF materials and thus on device lifetime had not

previously been recognized. As the majority of high-efficiency TADF materials contain cycloamino donors, our findings may be useful to the design of stable TADF molecules.

## Results

**The intrinsic stability of TADF molecules.** The chemical structures of the D–A molecules employed in our study are depicted in Fig. 1. The use of cycloamino donors is favored because they effectively suppress hazardous non-radiative relaxation and thus afford high quantum yields for fluorescence emission[25]. Various cycloamino D groups were employed, including carbazole (Cbz)[32], 10,11-dihydro-5H-dibenz[d,f]azepine (AZP)[30], 9,10-dihydro-9,9-dimethylacridine (DMAC)[33,34], phenoxazine (PXZ)[35–37], and phenothiazine (PTZ)[26–29]. We selected the cycloamino donors so as to vary their molecular parameters, such as their electrochemical potentials and conformation. Electrochemical potential is closely linked to the energy and lifetime of the fluorescent state, but the photophysical effects on stability of conformation are poorly understood despite the extensive study of the effects of conformation on the emergence of TADF or dual emissions[26,30,31,38–42]. This structural variation enabled us to identify the molecular factors critical to intrinsic stability. All the compounds were synthesized with Pd (II)-catalyzed C–N crosscoupling reactions, and their chemical structures and purity were determined with standard chemical techniques. The synthetic procedures and spectroscopic identification data of the compounds are summarized in Methods. Photophysical and electrochemical data for the compounds are compiled in Table 1, Supplementary Table 1 and Supplementary Figs. 1–3.

The intrinsic stabilities of the compounds were evaluated by using steady-state photolysis. Ar-saturated THF solutions of each compound (100 μM) were photoirradiated with broad-band light from a 300 W Xe lamp. Photolytic degradation was monitored by employing UV–vis absorption spectroscopy and high-performance liquid chromatography (HPLC) in combination with mass spectrometry (electrospray ionization (ESI), positive mode). As shown in Fig. 2a for AZP-TRZ, the absorbance of the ICT transition band at 363 nm (molar absorbance ($\varepsilon$) = $1.57 \times 10^4$ M$^{-1}$ cm$^{-1}$) decreases gradually during photolysis. An identical trend was also found for vacuum evaporated thick films of mCBP doped with 10 wt% AZP-TRZ (Supplementary Fig. 4). This disappearance is accompanied by the growth of weak absorption bands in the visible region that are eventually bleached. The emission bands in the photoluminescence spectra for the AZP-TRZ solution after 180 min photolysis are hypsochromically shifted from that of intact AZP-TRZ. These bands coincide with the fluorescence spectra of the independently synthesized fragment models, AZP-Ph and Ph-TRZ (Fig. 2b), which indicates the cleavage of the TRZ acceptor and the AZP donor.

Our mass spectrometry results provide supporting evidence of this cleavage. The degradation byproducts were separated with HPLC (reverse phase). The ESI mass analyses of each degradation product confirm the cleavage of the C–N and C–C bonds in AZP-TRZ (Fig. 2c). Similar degradation behaviors were observed for the other compounds (Supplementary Figs. 5–8). In all cases, the C–N bonds between the cycloamino donor and Ph-TRZ were found to be the degradation sites. Our quantum chemical calculations also show that the C–N bonds have the lowest bond dissociation energies (BDEs) (Supplementary Table 2). These findings indicate that the C–N bonds connecting the cycloamino donors and the phenyl bridges are the most vulnerable to degradation.

To analyze the degradation kinetics, the residual AZP-TRZ was quantified during the photolysis by employing HPLC

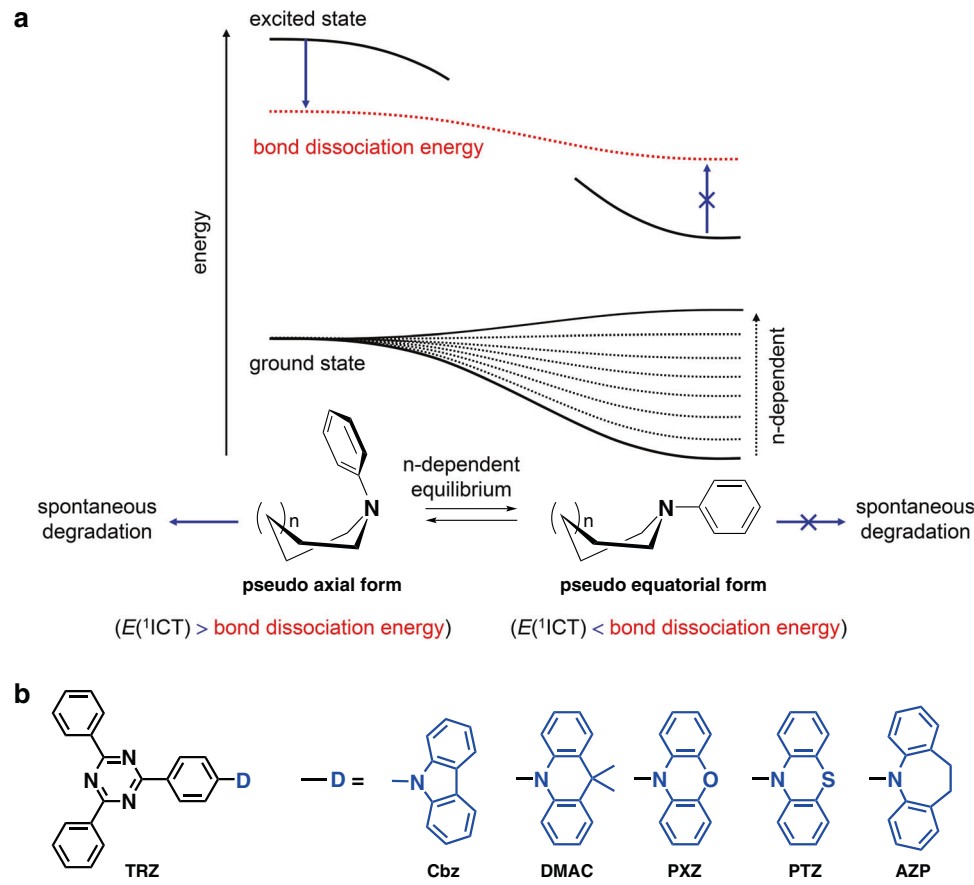

**Fig. 1 Intrinsic degradation of TADF dopants. a** Photophysical scheme for the conformation-dependent intrinsic degradation of TADF dopants. **b** Chemical structures of the TADF molecules employed in this study.

**Table 1 Photophysical and electrochemical data for the dyads.**

| | $\lambda_{abs}$ (nm)/$\varepsilon$ ($10^4$ $M^{-1}$ $cm^{-1}$)[a,b] | $\lambda_{em}$ (nm) [a,c] | PLQY[d,e] | $\tau_{PF}$ (ns)/ $\tau_{DF}$($\mu$s)[d,f] | $k_{r,PF}$ ($10^6$ $s^{-1}$)/ $k_{nr,PF}$ ($10^6$ $s^{-1}$)[g] | $k_{r,DF}$ ($10^6$ $s^{-1}$)/$k_{nr,DF}$ ($10^6$ $s^{-1}$)[h] | $E_{ox}$ (V vs SCE)[i] | $E_{red}$ (V vs SCE)[j] |
|---|---|---|---|---|---|---|---|---|
| Cbz-TRZ | 356/1.53 | 453 | 1.00 ± 0.01 | 4.2/-[k] | 240/0 | -/-[k] | 1.47 | −1.66 |
| AZP-TRZ | 364/5.09 | 494 | 0.20 ± 0.01 | 8.3/-[k] | 24/96 | -/-[k] | 1.33 | −1.56 |
| DMAC-TRZ | 380/0.26 | 552 | 0.70 ± 0.11 | 25/7.1 | 21/19 | 0.025/0.12 | 1.10 | −1.65 |
| PXZ-TRZ | 411/0.19 | 608 | 0.70 ± 0.03 | 20/1.6 | 15/35 | 0.25/0.37 | 0.88 | −1.66 |
| PTZ-TRZ | 356/1.48 | 435 | 0.40 ± 0.04 | 16/0.6 | 7.1/55 | 0.48/1.2 | 0.88 | −1.65 |

[a]50 $\mu$M in Ar-saturated EtOAc.
[b]Absorption peak wavelength.
[c]Photoluminescence peak wavelength. Excitation wavelengths = 363 nm (Cbz-TRZ), 372 nm (AZP-TRZ), 379 nm (DMAC-TRZ), 414 nm (PXZ-TRZ), and 359 nm (PTZ-TRZ).
[d]10 $\mu$M in Ar-saturated PhMe.
[e]Photoluminescence quantum yields determined relative to the 9,10-diphenylanthracene standard (PLQY = 1.0 in PhMe).
[f]Lifetime of the prompt ($\tau_{PF}$) and the delayed fluorescence ($\tau_{DF}$) obtained after pulsed laser excitation at 377 nm. Observation wavelength ($\lambda_{obs}$) = 417 nm (Cbz-TRZ), 461 nm (AZP-TRZ), 495 nm (DMAC-TRZ), 545 nm (PXZ-TRZ), and 561 nm (PTZ-TRZ).
[g]Radiative rate constant, $k_{r,PF}$ = PLQY$_{PF}$/$\tau_{PF}$ and $k_{r,DF}$ = PLQY$_{DF}$/$\tau_{DF}$.
[h]Non-radiative rate constant, $k_{nr,PF}$ = (1 − PLQY$_{PF}$)/$\tau_{PF}$ and $k_{nr,DF}$ = (1 − PLQY$_{DF}$)/$\tau_{DF}$. PLQY$_{PF}$ = 0.52 (DMAC-TRZ), 0.29 (PXZ-TRZ), 0.11 (PTZ-TRZ). PLQY$_{DF}$ = 0.18 (DMAC-TRZ), 0.41 (PXZ-TRZ), 0.29 (PTZ-TRZ).
[i]Oxidation potential.
[j]Reduction potential. The redox potentials were determined with cyclic and differential pulse voltammetry techniques for Ar-saturated THF containing 0.10 TBAPF$_6$ and 2.0 mM fluorophore. A Pt microdisc and a Pt wire were used as the working and counter electrodes, respectively. An Ag/AgNO$_3$ couple was employed as the pseudo-reference electrode.
[k]Not determined.

techniques with benzophenone as the internal standard (Fig. 2d). The apparent rate of the unimolecular photolytic degradation ($k_d$) was determined from the linear fit of the initial points of the AZP-TRZ concentration. Note that this linear fit does not necessarily indicate a zero-order kinetics. As shown in Fig. 2e, there are large differences in the apparent $k_d$ values. The largest $k_d$ value was found for AZP-TRZ ($k_d = 2400 \times 10^{-11}$ M s$^{-1}$), and these values decrease in the order Cbz-TRZ ($k_d = 720 \times 10^{-11}$ M s$^{-1}$) > DMAC-TRZ ($k_d = 170 \times 10^{-11}$ M s$^{-1}$) > PTZ-TRZ ($k_d = 38 \times 10^{-11}$ M s$^{-1}$) > PXZ-TRZ ($k_d = 4.8 \times 10^{-11}$ M s$^{-1}$). The quantum yields for the intrinsic degradations were determined by employing standard ferrioxalate actinometry (photon flux = $1.9 \times 10^{-7}$ einstein s$^{-1}$) and found to be 0.65 (AZP-TRZ), 0.18 (Cbz-TRZ), 0.044 (DMAC-TRZ), 0.010 (PTZ-TRZ), and 0.0013 (PXZ-TRZ). The variations in $k_d$ and the photochemical quantum yield are of several orders of magnitude, which indicates that the intrinsic stability of the compounds depends strongly on their chemical structures.

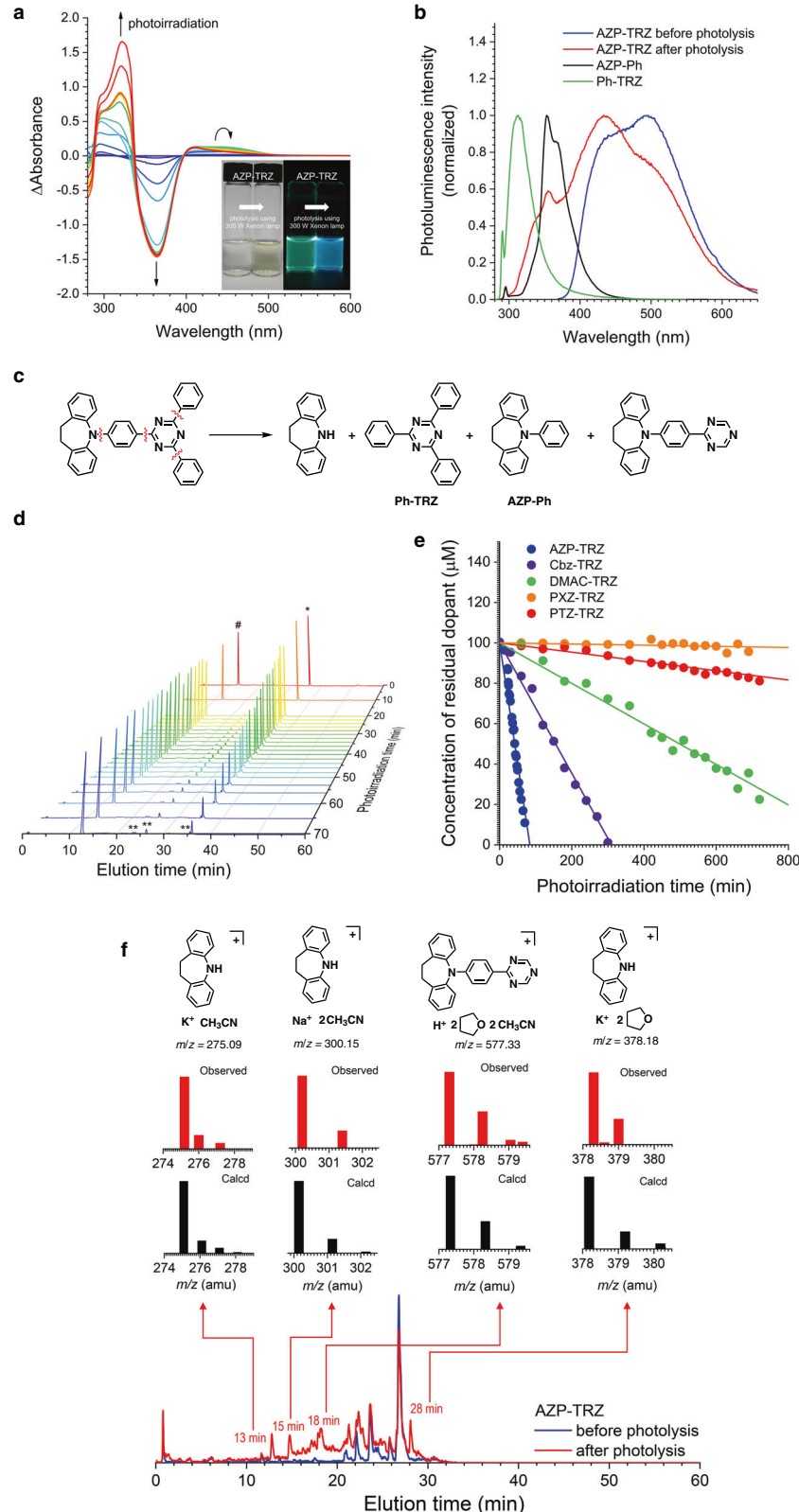

**Correlation of the operational lifetimes of TADF OLEDs and the intrinsic stability of TADF molecules.** In order to determine the relationship between the intrinsic stabilities of the compounds and the lifetimes of their devices, multi-layer OLEDs were fabricated by using the D–A dyads as dopants with the configuration ITO/DNTPD (60 nm)/BPBPA (20 nm)/PCzAC (10 nm)/mCBP:10 wt% dopant (30 nm)/DBF-Trz (5 nm)/ZADN (30 nm)/LiF:Al. In

this device, DNTPD (N,N′-diphenyl-N,N′-bis-[4-(phenyl-m-tolyl-amino-phenyl]-biphenyl-4,4′-diamine) comprises the hole-injection layer, BPBPA (N,N,N′N′-tetra[(1,1′-biphenyl)-4-yl]-(1,1′-biphenyl)-4,4′-diamine) comprises the hole-transporting layer, PCzAC (9,9-dimethyl-10-(9-phenyl-9H-carbazol-3-yl)-9,10-dihy-droacridine) comprises the electron-blocking layer, mCBP is the host, DBF-Trz (2,8-bis(4,6-diphenyl-1,3,5-triazin-2-yl)dibenzo[b,d]

**Fig. 2 Intrinsic degradation of AZP-TRZ. a** Changes in the UV–vis absorption spectra upon continuous photolysis of Ar-saturated THF containing 100 μM AZP-TRZ under illumination with broad-band light from a Xe lamp (300 W) for 180 min. **b** Comparison of the photoluminescence spectra of the AZP-TRZ solution (100 μM in Ar-saturated THF) before (blue) and after (red) photolysis and of the model compounds: AZP-Ph (black) and Ph-TRZ (green). See (**c**) for the chemical structures of AZP-Ph and Ph-TRZ. Excitation wavelengths: 295 nm (AZP-TRZ) and 288 nm (AZP-Ph and Ph-TRZ). Note that the photoluminescence spectra of AZP-TRZ do not vary upon photoexcitation at a wavelength of 288 nm. **c** Chemical equation for the degradation of AZP-TRZ. The red wavy lines indicate degradation sites. **d** Liquid chromatograms of Ar-saturated THF/$CH_3CN$ containing 100 μM AZP-TRZ during photolysis. The peaks marked with #, *, and ** correspond to the benzophenone internal standard, the intact AZP-TRZ, and the degradation byproducts of AZP-TRZ, respectively. The benzophenone internal standard (200 μM) was added after the photolysis. **e** Concentrations of the residual dopants during continuous photolysis. See text for details. **f** An overlay of the liquid chromatograms of AZP-TRZ before (blue) and after (red) photolysis. The partial mass spectra shown on top of the chromatograms are the calculated (black) and observed (red) isotope distributions of the indicated chromatographic peaks. The chemical structures above the mass spectra are the proposed structures corresponding to the calculated values.

furan) comprises the hole-blocking layer, and ZADN (2-(4-(9,10-di(naphthalen-2-yl)anthracen-2-yl)phenyl)-1-phenyl-1*H*-benzo[*d*]imidazole) comprises the electron-transporting layer. Devices including Cbz-TRZ were not tested due to unavailability of suitable host materials. The chemical structures and energy levels of the materials are shown in Fig. 3a. All the fabricated devices have very similar current density–voltage (*J–V*) profiles (Fig. 3b), which indicates that the injection, transport, and recombination of charge carriers do not vary from device to device. The highest-occupied molecular orbitals (HOMOs; 5.6–6.0 eV) of the dopant materials are quite different from that of the mCBP host, so hole carriers are probably transported through the host molecules. On the other hand, electrons are likely to be strongly trapped by the dopant because of the large LUMO gap between mCBP and the DBF-Trz layer (Fig. 3a). This strong electron trapping by the dopant is expected to dominate the charge carrier transport in the devices and thus dictates their current behaviors. In addition, the current behaviors indicate that the light emission in these devices originates from excitons produced by trap-assisted recombination. The electroluminescence properties of the devices, including their efficiencies, are summarized in Supplementary Table 3.

The operational stability of each device was evaluated by recording its relative luminance decay driven under a constant current density defined at an initial luminance of 200 cd m$^{-2}$. Figure 3c shows the luminance decay profiles as a function of operation time. The AZP-TRZ device is very short-lived, whereas the other devices are relatively long-lived. The operational lifetimes of the devices were estimated by determining $LT_{97}$, which is the time required for the luminance to reach 97% of its initial value. The $LT_{97}$ values are ordered as follows: 35 h (PXZ-TRZ) > 2.5 h (PTZ-TRZ) > 0.63 h (DMAC-TRZ) > 0.019 h (AZP-TRZ). An identical trend was found when all the devices were operated at an identical current density (9.25 mA cm$^{-2}$), which rules out the possibility that the variation in lifetimes originates in different charge carrier densities exclusively (Supplementary Fig. 9). Note particularly that $LT_{97}$ is linearly proportional to the intrinsic stability defined as $1/k_d$ (Fig. 3d).

As shown in Fig. 3e, the normalized electroluminescence spectra of the devices before and after operation are identical; thus the degradation byproducts make no contribution to the electroluminescence. PTZ-TRZ exhibited the electroluminescence spectrum bathochromically shifted from the photoluminescence spectrum due to polarity-dependent conformation changes. Therefore, the linearity of the relationship between $LT_{97}$ and $1/k_d$ provides firm evidence that the operational lifetimes of these devices are governed directly by the intrinsic stabilities of the dopants. The $LT_{97}$ values increase with the doping concentration of the dopant. However, the concentration effect becomes insignificant as the intrinsic stability of the dopant decreases (Supplementary Fig. 10). This observation indicates the decisive role of the individual excitonic state in the device stability. This rational is actually consistent with the weak adherence of the $LT_{97}$

trend to the offsets between the HOMO energies of the dopant and host materials (i.e., the extent of trap-assisted recombination of the dopant).

**Molecular factors that govern intrinsic stability**. The positive correlation between the operation lifetimes of the devices (i.e., $LT_{97}$) and the intrinsic stability of the materials (i.e., $1/k_d$) points to the importance of the structure–stability relationship. In order to identify the factors that govern the intrinsic stability, we attempted to correlate the excited-state properties of the compounds with $LT_{97}$ and $1/k_d$. It was found that $LT_{97}$ and $1/k_d$ depend weakly on the energy and unimolecular relaxation rates of the excitons (Supplementary Fig. 11).

This result prompted us to focus on the ground-state structures of the materials. Our X-ray single crystallographic analyses reveal that AZP-TRZ has a pseudo-axial form (Fig. 4a), in contrast to the crystal structure of PXZ-TRZ which has a pseudo-equatorial form (Fig. 4b). This conclusion for AZP-TRZ (AZP$^{ax}$-TRZ) is supported by our variable-temperature $^1$H NMR spectroscopy (298–373 K, 300 MHz, DMSO-$d_6$) results. The hydrogens that are *ortho* to AZP in the phenyl bridge experience a down-field shift at increased temperatures. This shift is ascribed to ring flip at elevated temperatures from the more stable pseudo-axial form to the less stable pseudo-equatorial form, in which the hydrogens are located at the vertical and lateral positions, respectively, of the benzene rings in AZP. This observation is also consistent with our simulated $^1$H NMR spectra (Supplementary Fig. 12).

The quantum chemical calculations performed at the CAM–B3LYP level of theory with the CPCM solvation model for THF support our structural assignments. The optimized geometry of AZP$^{ax}$-TRZ is predicted to be more stable than the pseudo-equatorial form (AZP$^{eq}$-TRZ) by 0.31 eV. The higher stability of AZP$^{ax}$-TRZ arises because there is less AZP ring strain in the pseudo-axial form than in the pseudo-equatorial form (Supplementary Fig. 13). In contrast, the pseudo-axial form of PXZ-TRZ (PXZ$^{ax}$-TRZ) is less stable than the pseudo-equatorial form (PXZ$^{eq}$-TRZ) by 0.20 eV. The higher stability of PXZ$^{eq}$-TRZ is demonstrated with the increased *s*-character of nitrogen in PXZ in this form (Supplementary Fig. 14). The validity of our results is supported by the close resemblance of the experimental (50 μM in THF, 298 K) and simulated (TD–CAM–B3LYP) UV–vis absorption spectra (Supplementary Fig. 15). Finally, the strong positive solvatochromism of the UV–vis absorption spectra of AZP-TRZ is consistent with the observation that the dipole moment of AZP$^{ax}$-TRZ (3.7114 Debye) is greater than that of AZP$^{eq}$-TRZ (1.1223 Debye) (Supplementary Fig. 16).

Our structural, spectroscopic, and quantum chemical results collectively indicate that the dyads are present as two different conformers with ratios that depend on the structure of the cycloamino donor. AZP-TRZ has a seven-membered cycloamino donor and exists exclusively in a pseudo-axial form. In contrast, the other dyads have six-membered cycloamino donors and are

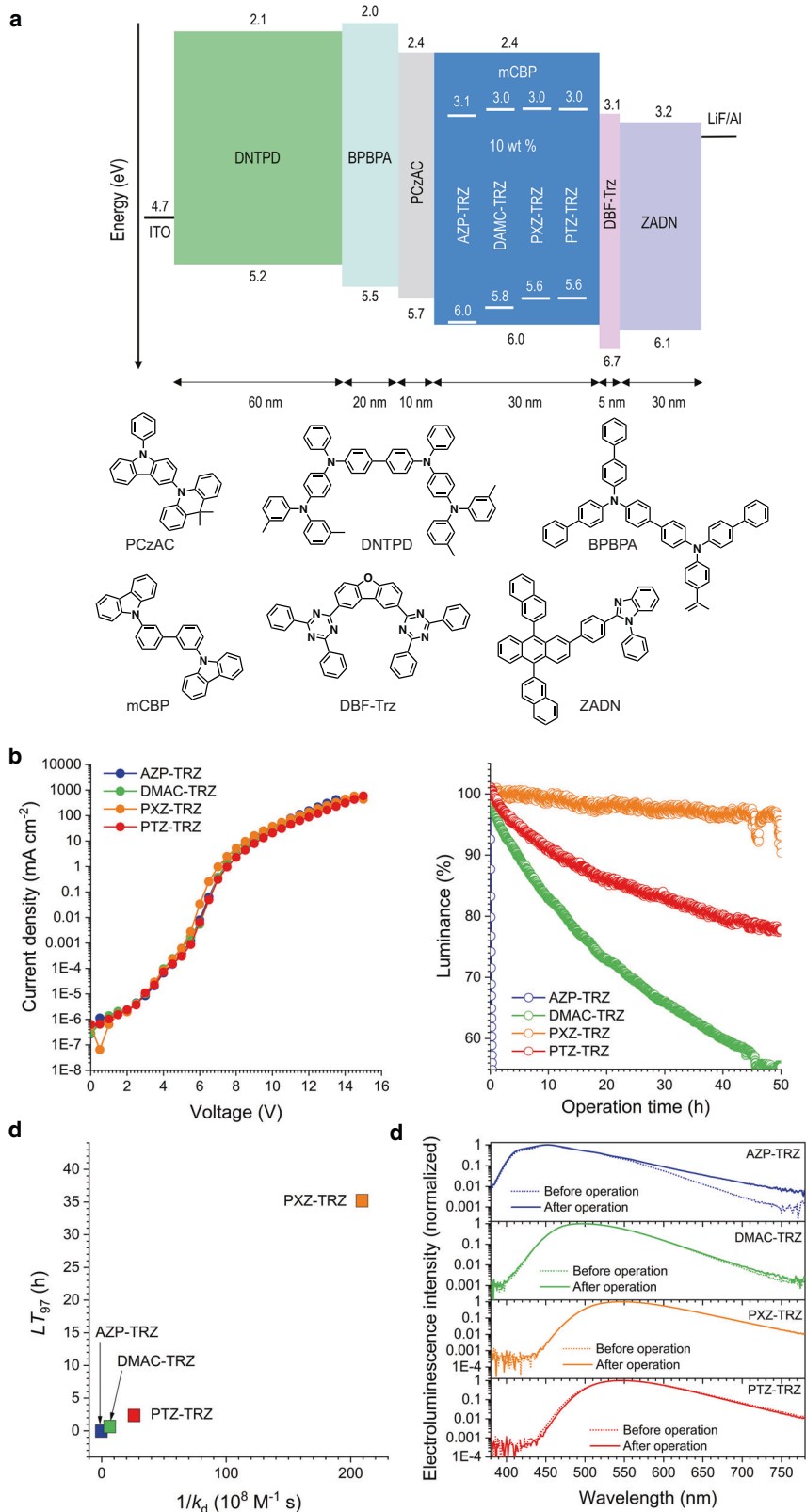

**Fig. 3 Electroluminescence performances. a** A schematic representation of the configuration of the devices and the chemical structures of the materials. **b** Plots of the current density as a function of the applied voltage for the devices. **c** Changes in the % luminance during operation of the devices driven at constant current densities defined at an initial luminance of 200 cd m$^{-2}$: 1.07 mA cm$^{-2}$ (AZP-TRZ), 0.93 mA cm$^{-2}$ (DMAC-TRZ), 0.43 mA cm$^{-2}$ (PXZ-TRZ), and 0.51 mA cm$^{-2}$ (PTZ-TRZ). **d** A positive correlation between the device lifetime ($LT_{97}$) and the intrinsic stability of the compounds ($1/k_d$). **e** Electroluminescence spectra of devices before (dotted lines) and after (solid lines) operation.

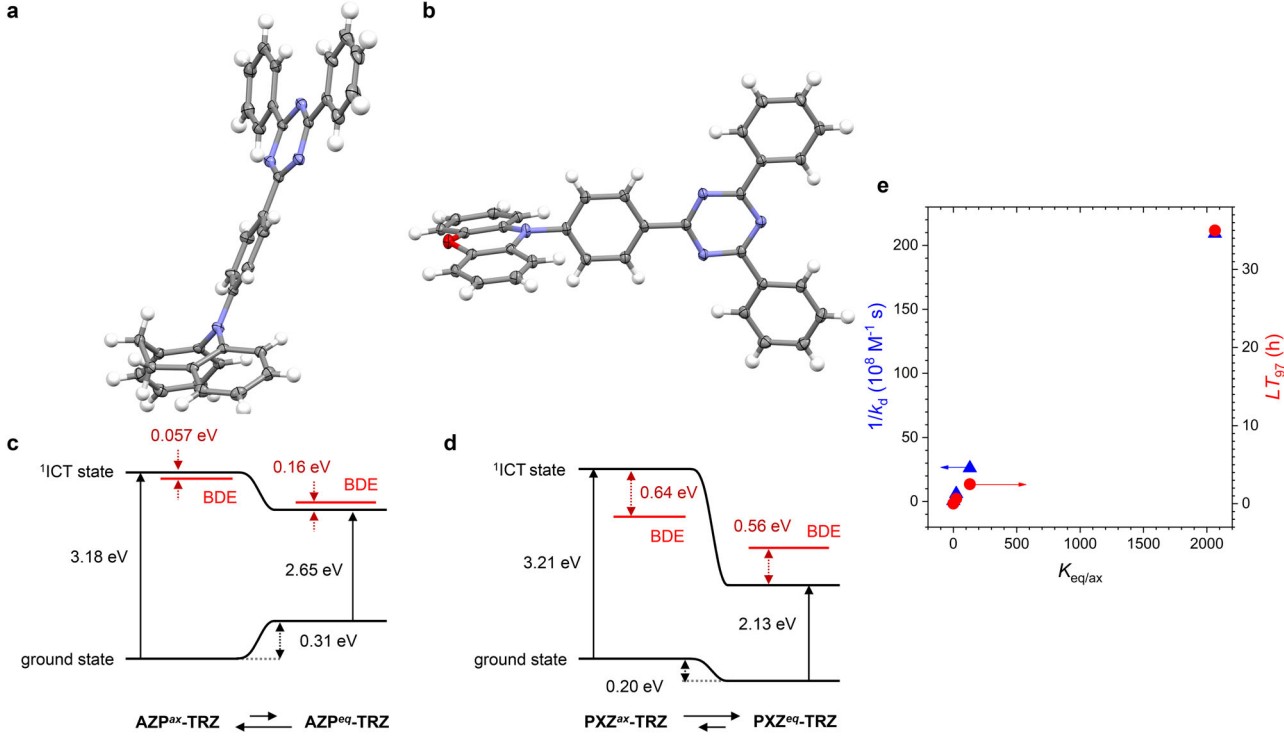

**Fig. 4 Conformation dependence of degradation. a, b** Oak Ridge thermal ellipsoid plots of the X-ray crystal structures of (**a**) AZP-TRZ and (**b**) PXZ-TRZ, drawn at 30% probability. The structure of PXZ-TRZ was obtained from ref. [36]. **c, d** Schematic representations of the energy levels of the ground and $^1$ICT transition states of the pseudo-axial and pseudo-equatorial forms of (**c**) AZP-TRZ and (**d**) PXZ-TRZ. The red bars indicate the bond dissociation energies relative to the $^1$ICT transition states. **e** Elucidation of the molecular parameters that govern the intrinsic stabilities of the molecules (i.e., $1/k_d$; blue triangles) and the operation lifetimes of the devices (i.e., $LT_{97}$; red circles). Plots of $1/k_d$ and $LT_{97}$ as functions of the Boltzmann factor of the pseudo-equatorial form and the pseudo-axial form $K_{eq/ax}$, $K_{eq/ax} = \exp(-\Delta E_{eq-ax}/k_B T)$, estimated at 298 K. $\Delta E_{eq-ax}$ = (ground-state energy of the pseudo-equatorial form) − (ground-state energy of the pseudo-axial form).

**Table 2 Quantum chemical predictions of the stabilities of the pseudo-equatorial and the pseudo-axial forms[a].**

| | $\Delta E_{eq-ax}$ (eV)[b] | $\Delta H_{C-N}$ (eV)[c] | | $^1$ICT (eV)[d] | | BDE (eV)[e] | |
|---|---|---|---|---|---|---|---|
| | | Pseudo-equatorial form | Pseudo-axial form | Pseudo-equatorial form | Pseudo-axial form | Pseudo-equatorial form | Pseudo-axial form |
| AZP-TRZ | 0.314 | 0.16 | −0.06 | 2.65 | 3.18 | 2.81 | 3.12 |
| DMAC-TRZ | −0.0803 | 0.44 | −0.28 | 2.51 | 3.15 | 2.95 | 2.87 |
| PXZ-TRZ | −0.125 | 0.56 | −0.64 | 2.13 | 3.21 | 2.69 | 2.57 |
| PTZ-TRZ | −0.196 | 0.31 | −0.50 | 2.41 | 3.24 | 2.72 | 2.74 |

[a]A full list of the quantum chemical calculation results, including the C–N bond lengths, the bond dissociation energies of C–N and C–C, the exchange energies of the intramolecular charge-transfer (ICT) transition states (i.e., $^1$ICT–$^3$ICT), and the energy differences between the $^1$ICT transition state and the local excitation state is compiled in Supplementary Table 2.
[b]$\Delta E_{eq-ax}$ = (the ground-state energy of the pseudo-equatorial form) − (the ground-state energy of the pseudo-axial form). A negative value indicates that the pseudo-equatorial form is more stable than the pseudo-axial form.
[c]$\Delta H_{C-N}$ = (the bond dissociation energy of C–N) − (the $^1$ICT transition state energy). Negative and positive values correspond to exothermic and endothermic bond dissociation, respectively.
[d]Energy of transition from the ground state to the $^1$ICT transition state.
[e]Bond dissociation energy of the most vulnerable C–N.

present in varying ratios of the pseudo-axial and pseudo-equatorial forms, with the latter as the major conformer. It should be noted that Cbz-TRZ possesses a rigid cycloamino donor, excluding the pseudo-axial and pseudo-equatorial conformerization to be responsible for the moderate instability. We hypothesize that the formation of an intramolecular charge-separated species might lead to cleavage of the C–N bond in Cbz-TRZ (see Supplementary Fig. 17). The energy differences ($\Delta E_{eq-ax}$) between the pseudo-equatorial and pseudo-axial forms of the dyads are shown in Table 2. The equilibrium constant in each case for interconversion of the pseudo-equatorial and pseudo-axial forms ($K_{eq/ax}$) was estimated by using the energy

difference and the equation $K_{eq/ax} = \exp(-\Delta E_{eq-ax}/k_B T)$, where $k_B$ and $T$ are the Boltzmann constant and room temperature (298 K), respectively. The $K_{eq/ax}$ values are $4.9 \times 10^{-6}$ for AZP-TRZ, 23 for DMAC-TRZ, 130 for PTZ-TRZ, and 2060 for PXZ-TRZ.

The striking influence of the conformeric heterogeneity is illustrated in Fig. 4c–d. The BDE for the vulnerable C–N bond is smaller than the $^1$ICT transition energy in AZP$^{ax}$-TRZ, but is greater than the $^1$ICT transition energy in AZP$^{eq}$-TRZ. This observation suggests that exoergic bond dissociation in the excitonic state of AZP$^{ax}$-TRZ has a high probability, whereas the dissociation is thermodynamically disallowed in the excitonic state of AZP$^{eq}$-TRZ. The thermodynamic driving forces for the

C–N bond cleavages ($\Delta H_{C-N}$) in AZP$^{ax}$-TRZ and AZP$^{eq}$-TRZ, as quantified by the equation $\Delta H_{C-N} = BDE - E(^1ICT)$, are −0.057 and 0.16 eV, respectively. A similar trend is evident in the $\Delta H_{C-N}$ values of −0.64 and 0.56 eV for PXZ$^{ax}$-TRZ and PXZ$^{eq}$-TRZ, respectively. The $\Delta H_{C-N}$ values of the conformers of the compounds are tabulated in Table 2. In all cases, the spontaneous degradation (i.e., $\Delta H_{C-N} < 0$) of the pseudo-axial conformer is predicted. Our results reveal that the magnitude of BDE cannot be used as a figure of merit of intrinsic stability. Our mechanistic studies establish that stability is actually governed by the conformeric heterogeneity (i.e., $K_{eq/ax}$). In particular, stability improves with the increased abundance of the pseudo-equatorial conformer, which has a positive $\Delta H_{C-N}$ (i.e., C–N bond scission is thermodynamically forbidden). This prediction is consistent with the excellent linearity between $K_{eq/ax}$ and both the intrinsic stability (i.e., $1/k_d$) and the operation lifetime ($LT_{97}$) (Fig. 4e).

In addition to the degradation thermicity, the conformeric heterogeneity was intimately linked to the level of dark triplet excitons. The absence of delayed fluorescence from AZP-TRZ having the pseudo-axial form as the major conformer suggested a conformation dependence on reverse intersystem crossing (rISC). Our quantum chemical calculations at the level of TD–CAM–B3LYP revealed the contrasting conformation dependence of an exchange energy between the singlet ($^1ICT$) and triplet ($^3ICT$) intramolecular charge-transfer transition states ($\Delta E_{1ICT-3ICT}$). Contrary to the pseudo-equatorial forms having small $\Delta E_{1ICT-3ICT}$ of 0.01–0.04 eV, the pseudo-axial forms were calculated to have one order of magnitude greater $\Delta E_{1ICT-3ICT}$ (0.13–0.15 eV) (Supplementary Table 1). This prediction accords with the quantum yield for rISC ($\Phi_{rISC}$). $\Phi_{rISC}$ increases with $K_{eq/ax}$ (Supplementary Fig. 11). These experimental and computational results coherently point to the conformeric heterogeneity as the key factor that control triplet exciton levels. Since long-lived dark triplet excitons are vulnerable to degradation, TADF molecules with greater pseudo-equatorial conformers would improve the operational stability of devices. This hypothesis is evidenced by our electroluminescence results (Fig. 4e), demonstrating the validity of our finding.

## Discussion

The high electroluminescence performances of TADF OLEDs promise a range of advances in display technologies. One challenge for TADF OLEDs that needs to be addressed is their insufficient operational lifetimes. Although several approaches have been pursued based on device engineering, molecular strategies for the creation of robust TADF materials remain underdeveloped. This strategy is of enormous importance because the operational instability of devices is due principally to the intrinsic degradation of TADF materials. We have determined the molecular factors that govern the intrinsic stability of a series of archetypal D–A TADF molecules. We performed photolysis experiments to quantify the stability of the molecules, which can be defined as the inverse of the rate of exciton-localized degradation ($1/k_d$). It was found that the operation lifetime ($LT_{97}$) of OLEDs is directly proportional to the intrinsic stability of the TADF molecules. Unexpectedly, both $LT_{97}$ and $1/k_d$ do not correlate with the energy and relaxation rates of the electronic state producing TADF. We examined the effects of structure on $LT_{97}$ and $1/k_d$, and found that both are governed by the conformeric heterogeneity of the cycloamino donor units: specifically, exoergic bond dissociation occurs from the excitonic state of the pseudo-axial conformer. Bond rupture is forbidden in the pseudo-equatorial conformer. The pseudo-axial and pseudo-equatorial forms of the TADF molecules are present in varying ratios of relative abundance that depends on the chemical structure of the cycloamino donor. We found that the relative abundance of the two conformers for a TADF molecule dictates its intrinsic stability. This unprecedented finding provides a unique guideline for the design of TADF molecules: TADF compounds with prevalent pseudo-equatorial conformers will exhibit longer device lifetimes. Since the majority of high-efficiency TADF materials involve cycloamino donors, our research will be useful for improving the operational stability of TADF OLEDs.

## Methods

**Materials**. Commercially available chemicals, including 2-(4-bromophenyl)-4,6-diphenyl-1,3,5-triazine (TCI), 10,11-dihydro-5H-dibenz[b,f]azepine (TCI), 9,10-dihydro-9,9-dimethylacridine (TCI), phenoxazine (Alfa-Aesar), phenothiazine (TCI), potassium carbonate (Sigma-Aldrich), palladium(II) acetate (Sigma-Aldrich), tri-tert-butylphosphine (Sigma-Aldrich), and DMAC-Ph (TCI) were used without further purification, unless otherwise stated. All glassware, magnetic stir bars, syringes, and needles were dried in a convection oven at 120 °C. Reactions were monitored by using thin-layer chromatography (TLC). Commercial TLC plates (silica gel 254, Merck Co.) were developed, and the spots were visualized under UV irradiation at wavelengths of 254 or 365 nm. Silica gel column chromatography was performed with silica gel 60G (particle size 0.040–0.063 mm, Merck Co.). $^1H$ NMR (300 MHz) and $^{13}C\{^1H\}$ NMR (126 MHz) spectra were collected with Bruker, AVANCE III 300 and 500 spectrometers, respectively. Chemical shifts were referenced to the peaks due to a residual solvent. Low- and high-resolution mass spectra were recorded by using an Agilent Technologies 6890 series mass spectrometer.

**Synthesis of Cbz-TRZ**. 2-(4-Bromophenyl)-4,6-diphenyl-1,3,5-triazine (5.00 g, 12.9 mmol), carbazole (2.37 g, 14.2 mmol), and potassium carbonate (5.34 g, 38.6 mmol) were dissolved in toluene (120 mL) in a 250 mL two-necked round-bottom flask equipped with a magnetic stir bar. Palladium(II) acetate (0.0867 g, 0.380 mmol) and tri-tert-butylphosphine (0.287 g, 1.42 mmol) dissolved in toluene (20 mL) was delivered to a stirred reaction mixture under an Ar atmosphere. The solution was stirred at 110 °C for 37 h under an Ar atmosphere. Greenish brown precipates were formed upon cooling the solution to room temperature, which were filtered and washed thoroughly with water. The crude product was recovered by dissolving in $CH_2Cl_2$, and the solution was concentrated under a reduced pressure. Silica gel column purification was performed with the eluent of $CH_2Cl_2$:hexane = 1:4 (v/v). Yellow powders were obtained in a 13% yield. $R_f = 0.27$ ($CH_2Cl_2$:hexane = 1:2, v/v). $^1H$ NMR (300 MHz, CDCl$_3$) δ (ppm): 7.34 (m, 2H), 7.45 (m, 2H), 7.61 (m, 8H), 7.83 (d, $J = 8.7$ Hz, 2H), 8.18 (d, $J = 7.8$ Hz, 2H), 8.83 (m, 4H), 9.03 (d, $J = 8.7$ Hz, 2H). $^{13}C\{^1H\}$ NMR (126 MHz, CD$_2$Cl$_2$) δ (ppm): 27.2, 30.3, 32.5, 110.5, 120.9, 124.2, 126.7, 127.3, 129.3, 129.5, 131.1, 133.3, 135.5, 136.7, 141.0, 142.2, 171.6, 172.4. HR MS (FAB, m-NBA): Calcd for $C_{33}H_{23}N_4$ ([M + H]$^+$), 475.1923; found: 475.1927.

**Synthesis of AZP-TRZ**. AZP-TRZ was prepared following a procedure identical to that used in the synthesis of Cbz-TRZ, except that 10,11-dihydro-5H-dibenz[b,f] azepine was used in place of carbazole. Silica gel column purification with the eluent $CH_2Cl_2$:hexane = 1:2 (v/v) afforded a yellow powder in a yield of 16%. $R_f = 0.33$ ($CH_2Cl_2$:hexane = 1:2, v/v). $^1H$ NMR (300 MHz, CDCl$_3$) δ (ppm): 3.04 (s, 4H), 6.73 (d, $J = 9.3$ Hz, 2H), 7.31 (m, 6H), 7.46 (m, 2H), 7.56 (m, 6H), 8.54 (d, $J = 9.0$ Hz, 2H), 8.72 (m, 4H). $^{13}C\{^1H\}$ NMR (126 MHz, CD$_2$Cl$_2$) δ (ppm): 30.3, 31.2, 112.9, 125.7, 127.8, 128.2, 129.1, 129.3, 129.4, 129.9, 130.9, 131.7, 132.7, 137.2, 138.6, 143.4, 153.2, 171.6, 171.8. HR MS (FAB, m-NBA): Calcd for $C_{35}H_{27}N_4$ ([M + H]$^+$), 503.2236; found: 503.2232.

**Synthesis of DMAC-TRZ**. DMAC-TRZ was prepared following a procedure identical to that used in the synthesis of Cbz-TRZ, except that 9,10-dihydro-9,9-dimethylacridine was used in place of carbazole. Silica gel column purification with the eluent of $CH_2Cl_2$:hexane = 1:4 (v/v) afforded a yellow powder in a yield of 48%. $R_f = 0.20$ ($CH_2Cl_2$:hexane = 1:2, v/v). $^1H$ NMR (300 MHz, CDCl$_3$) δ (ppm): 1.73 (s, 6H), 6.38 (m, 2H), 6.98 (m, 4H), 7.50 (m, 2H), 7.61 (m, 8H), 8.82 (m, 4H), 9.03 (d, $J = 6.6$ Hz, 2H). $^{13}C\{^1H\}$ NMR (126 MHz, CD$_2$Cl$_2$) δ (ppm): 30.2, 114.0, 116.0, 122.1, 123.9, 129.3, 129.5, 131.6, 132.2, 133.3, 134.6, 136.6, 136.9, 143.6, 144.5, 171.5, 172.4. HR MS (FAB, m-NBA): Calcd for $C_{36}H_{29}N_4$ ([M + H]$^+$), 517.2392; found: 517.2386.

**Synthesis of PXZ-TRZ**. PXZ-TRZ was prepared following a procedure identical to that used in the synthesis of Cbz-TRZ, except that phenoxazine (1.56 g, 8.50 mmol) was used in place of carbazole. Silica gel column purification with the eluent $CH_2Cl_2$:hexane = 1:4 (v/v) gave a yellow powder in a 45% yield. $R_f = 0.20$ ($CH_2Cl_2$: hexane = 1:2, v/v). $^1H$ NMR (300 MHz, CDCl$_3$) δ (ppm): 6.05 (dd, $J = 7.8$, 1.5 Hz, 2H), 6.62 (m, 2H), 6.68 (m, 2H), 6.74 (dd, $J = 7.8$, 1.8 Hz, 2H), 7.60 (m, 8H), 8.81 (d, $J = 8.4$ Hz, 4H), 9.00 (d, $J = 4.4$ Hz, 2H). $^{13}C\{^1H\}$ NMR (126 MHz, CD$_2$Cl$_2$) δ

(ppm): 27.8, 30.3, 121.3, 124.7, 124.9, 126.7, 127.7, 128.2, 129.2, 129.4, 11.5, 133.1, 133.5, 136.8, 143.5, 147.5, 171.6, 172.2. HR MS (FAB, $m$-NBA): Calcd for $C_{33}H_{22}N_4O$ ([M]$^+$), 490.1800; found: 490.1794.

**Synthesis of PTZ-TRZ**. PTZ-TRZ was prepared following a procedure identical to that used in the synthesis of Cbz-TRZ, except that phenothiazine was used in place of carbazole. Silica gel column purification with the eluent $CH_2Cl_2$:hexane = 1:4 (v/v) afforded a yellow powder in a 45% yield. $R_f$ = 0.30 ($CH_2Cl_2$:hexane = 1:2, v/v). $^1H$ NMR (300 MHz, CDCl$_3$) $\delta$ (ppm): 6.70 (d, $J$ = 3.9 Hz, 2H), 6.96 (m, 2H), 7.04 (m, 2H), 7.20 (d, $J$ = 7.2 Hz, 2H), 7.48 (d, $J$ = 8.7 Hz, 2H), 7.59 (m, 6H), 8.78 (d, $J$ = 8.1 Hz, 4H), 8.88 (d, $J$ = 9.0 Hz, 2H). $^{13}C\{^1H\}$ NMR (126 MHz, CD$_2$Cl$_2$) $\delta$ (ppm): 27.8, 30.3, 121.3, 124.7, 124.9, 126.7, 127.7, 128.2, 129.2, 129.4, 131.5, 133.1, 133.5, 136.8, 143.5, 147.5, 171.6, 172.2. HR MS (FAB, $m$-NBA): Calcd for $C_{33}H_{22}N_4S$ ([M]$^+$), 506.1570; found: 506.1565.

**Synthesis of AZP-Ph**. Bromobenzene (0.20 g, 1.3 mmol), 10,11-dihydro-5$H$-dibenz[$b$,$f$]azepine (0.27 g, 1.4 mmol) and potassium carbonate (0.53 g, 3.8 mmol) were added to a 50 mL two-necked round-bottom flask equipped with a magnetic stir bar. Fifteen milliliters of toluene was added to the reaction mixture under an Ar atmosphere. Palladium(II) acetate (8.6 mg, 0.38 mmol) and tri-$tert$-butylphosphine (28 mg, 0.14 mmol) were dissolved in toluene (2 mL), and transferred to the stirred solution under an Ar atmosphere. The solution was heated at 110 °C for 6 days. Dark purple precipitates formed upon cooling, which were filtered and washed thoroughly with water. The precipitates were collected by dissolving in $CH_2Cl_2$. Silica gel column purification was performed with the eluent $CH_2Cl_2$:hexane = 1:4 (v/v) to afford a gray powder in a 32% yield. $R_f$ = 0.37 ($CH_2Cl_2$:hexane = 1:4, v/v). $^1H$ NMR (300 MHz, CDCl$_3$) $\delta$ (ppm): 2.99 (s, 4H), 6.57 (m, 2H), 6.70 (m, 1H), 7.10 (m, 2H), 7.22 (m, 5H), 7.27 (m, 1H), 7.40 (m, 2H). $^{13}C\{^1H\}$ NMR (126 MHz, CD$_2$Cl$_2$) $\delta$ (ppm): 31.3, 113.1, 118.0, 127.6, 127.7, 129.3, 130.6, 131.5, 139.0, 144.1, 149.7. HR MS (FAB, $m$-NBA): Calcd for $C_{20}H_{17}N$ ([M]$^+$), 271.1356; found: 271.1361. Anal. Calcd for $C_{20}H_{17}N$: C, 88.52; H, 6.31; N, 5.16. Found: C, 88.73; H, 6.21; N, 5.39.

**Synthesis of PXZ-Ph**. PXZ-Ph was prepared with a procedure identical to that used in the synthesis of AZP-Ph, except that phenoxazine was used in place of 10,11-dihydro-5$H$-dibenz[$b$,$f$]azepine. Silica gel column purification was performed with the eluent $CH_2Cl_2$:hexane = 1:4 (v/v) to produce a yellowish-white powder in a 58% yield. $R_f$ = 0.37 ($CH_2Cl_2$:hexane = 1:4, v/v). $^1H$ NMR (300 MHz, CDCl$_3$) $\delta$ (ppm): 5.90 (d, $J$ = 4.5 Hz, 2H), 6.62 (m, 6H), 7.34 (d, $J$ = 7.5 Hz, 2H), 7.47 (t, $J$ = 7.5 Hz, 1H), 7.60 (m, 2H). $^{13}C\{^1H\}$ NMR (126 MHz, CD$_2$Cl$_2$) $\delta$ (ppm):113.8, 115.8, 121.8, 123.8, 129.1, 131.3, 131.6, 135.0, 139.5, 144.5. HR MS (FAB, $m$-NBA): Calcd for $C_{18}H_{13}NO$ ([M]$^+$), 259.0996; found: 259.0997. Anal. Calcd for $C_{18}H_{13}NO$: C, 83.37; H, 5.05; N, 5.40; O, 6.17. Found: C, 83.24; H, 5.02; N, 5.57; O, 6.16.

**Synthesis of PTZ-Ph**. PTZ-Ph was prepared with a procedure identical to that used in the synthesis of AZP-Ph, except that phenothiazine (0.28 g, 1.4 mmol) was used in place of 10,11-dihydro-5$H$-dibenzo[$b$,$f$]azepine. Silica gel column purification was performed with the eluent $CH_2Cl_2$:hexane = 1:4 (v/v) to afford a yellowish-white powder in a 65% yield. $R_f$ = 0.37 ($CH_2Cl_2$:hexane = 1:4, v/v). $^1H$ NMR (300 MHz, CDCl$_3$) $\delta$ (ppm): 6.20 (d, $J$ = 6.9 Hz, 2H), 6.96 (d, $J$ = 6.3 Hz, 4H), 7.02 (m, 2H), 7.40 (d, $J$ = 7.2 Hz, 2H), 7.48 (m, 1H), 7.60 (t, $J$ = 7.5 Hz, 2H). $^{13}C\{^1H\}$ NMR (126 MHz, CD$_2$Cl$_2$) $\delta$ (ppm): 116.7, 121.0, 123.1, 127.4, 128.7, 131.3, 144.3. HR MS (FAB, $m$-NBA): Calcd for $C_{18}H_{13}NS$ ([M]$^+$), 275.0770; found: 275.0769. Anal. Calcd for $C_{18}H_{13}NS$: C, 78.51; H, 4.76; N, 5.09; S, 11.64. Found: C, 78.66; H, 4.75; N, 5.21; S, 11.35.

**X-ray crystallography**. Crystals suitable for X-ray crystallography were obtained by allowing diffusion of ether vapor onto a $CH_2Cl_2$ solution of 10 mM AZP-TRZ. Single crystals were picked from the solution by using a nylon loop (Hampton Research Co.) at room temperature. Data collection for a single crystal was conducted on a Bruker, SMART CCD diffractometer equipped with a graphite-monochromated Mo K$\alpha$ ($\lambda$ = 0.71073 Å) radiation source under a nitrogen cold stream (223 K). Data collection and integration were performed with a Bruker, SAINT program. Semi-empirical absorption corrections based on equivalent reflections were applied with the Bruker SADABS program. Structures were obtained by using direct methods and refined using a full-matrix least-squares method on F2 by using SHELXL97. All non-hydrogen atoms were refined anisotropically. Hydrogen atoms were added to their geometrically ideal positions. Crystal data for AZP-TRZ: $C_{35}H_{26}N_4$, monoclinic, P2$_1$/c, Z = 8, a = 19.8371(6), b = 9.7227(3), c = 26.6431(9) Å, $\alpha$ = 90°, $\beta$ = 91.6354(12)°, $\gamma$ = 90°, V = 5136.6(3) Å$^3$, $\mu$ = 0.077 mm$^{-1}$, $\rho_{calcd}$ = 1.300 g cm$^{-3}$, $R_1$ = 0.0481, $wR_2$ = 0.0889 for 196332 unique reflections, 703 variables. The crystallographic data for AZP-TRZ are summarized in Supplementary Table 4. Supplementary Tables 5 and 6 list selected bond distances and angles, respectively.

**Steady-state UV–vis absorption measurements**. UV–vis absorption spectra were collected on an Agilent, Cary 300 spectrophotometer at 298 K. Stock solutions of the dyads with a concentration of 10 mM were prepared in ethyl acetate. Sample solutions were prepared prior to measurements by diluting the stock solution in ethyl acetate to concentrations of 10 or 100 μM. 3.0 mL of each solution was delivered into a quartz cell (Hellma, beam path length = 1.0 cm).

**Steady-state fluorescence measurements**. Fluorescence spectra were obtained by using a Photon Technology International, Quanta Master 400 scanning spectrofluorometer at 298 or 80 K. Ten or 100 μM solutions were used in these measurements, unless otherwise stated. The excitation wavelengths were 363 nm (Cbz-TRZ), 371 nm (AZP-TRZ), 387 nm (DMAC-TRZ), 420 nm (PXZ-TRZ), and 367 nm (PTZ-TRZ). All operations were controlled with the FelixGX software provided by the manufacturer.

**Fluorescence lifetime measurements**. Ten micromolar solutions in Ar-saturated toluene were used for the determination of the fluorescence lifetimes ($\tau_{obs}$). Fluorescence decay traces were acquired with the time-correlated single-photon-counting (TCSPC) technique by using a PicoQuant, FluoTime 200 instrument after picosecond pulsed laser excitation. A 377 nm diode laser (PicoQuant, Germany) was used as the excitation source. The burst mode was used to monitor delayed fluorescence. The short-lived prompt fluorescence signals were re-examined in normal mode operation. The fluorescence signals at the emission peak wavelengths of the compounds were obtained with an automated motorized monochromator: Cbz-TRZ, 417 nm; AZP-TRZ, 461 nm; DMAC-TRZ, 495 nm; PXZ-TRZ, 545 nm; PTZ-TRZ, 561 nm. Fluorescence decay profiles were analyzed (OriginPro 8.0, OriginLab) by using a biexponential decay model.

**Solvatochromism**. The fluorescence spectra of 50 μM solutions were acquired, and the wavenumber of the peak wavelength of each spectrum was calculated. The Lippert–Mataga plot was constructed as a function of the polarity parameter $f$; $f$ = $((\varepsilon - 1)/(2\varepsilon + 1)) - ((n^2 - 1)/(2n^2 + 1))$ ($\varepsilon$, dielectric constant; $n$, refractive index). $f$: toluene, 0.0131; 1,4-dioxane, 0.0205; ethyl acetate, 0.200; dichloromethane, 0.218; 1,2-dichloroethane, 0.220.

**Determination of photoluminescence quantum yields**. The relative photoluminescence quantum yield (PLQY) was determined for each toluene solution, by using the equation PLQY = PLQY$_{ref}$ × ($I/I_{ref}$) × ($A_{ref}/A$) × ($n/n_{ref}$)$^2$, where $A$, $I$, and $n$ are the absorbance at the excitation wavelength, the integrated photoluminescence intensity, and the refractive index of the solvent, respectively. 9,10-Diphenylanthracene (PLQY$_{ref}$ = 1.00, toluene; $\lambda_{ex}$ = 366 nm) was used as the external reference. Each 10 μM sample and the reference solutions were thoroughly deaerated prior to the measurements. Photoluminescence spectra were collected at 298 K in the emission range 400–700 nm. The spectra were integrated by employing OriginLab, OriginPro 2018 software.

**Electrochemical characterization**. Cyclic and differential pulse voltammetry experiments were carried out by using a CH Instruments, CHI630 B potentiostat with a three-electrode cell assembly. A Pt wire and a Pt microdisc were used as the counter and working electrodes, respectively. An Ag/AgNO$_3$ couple was used as the pseudo-reference electrode. Measurements were carried out in Ar-saturated THF (2.0 mL) by using 0.10 M tetra-$n$-butylammonium hexafluorophosphate (Bu$_4$NPF$_6$) as the supporting electrolyte at scan rates of 100 mV s$^{-1}$ (cyclic voltammetry) and 4.0 mV s$^{-1}$ (differential pulse voltammetry). A ferrocenium/ferrocene couple was employed as the external reference.

**Quantum chemical calculations**. Quantum chemical calculations were carried out by using a Gaussian 09 program. Geometry optimization was performed by using the Coulomb attenuated method (CAM)-Becke's three parameter B3LYP exchange-correlation functional, and the 6−311 + G(d,p) basis set. The input geometries for the pseudo-axial and pseudo-equatorial forms were found to yield optimized geometries in the pseudo-axial and pseudo-equatorial forms, respectively. Frequency calculations were subsequently performed to assess the stability of the convergence. Natural bond analyses (NBO) were performed for the ground states. TD–CAM–B3LYP calculations were performed to simulate the electronic transition energies. GaussSum was employed for the simulation of the predicted electronic absorption spectra. Bond dissociation energies (BDEs) were calculated from the relationship BDE = $E(D\bullet) + E(\bullet Ph\text{-}TRZ) - E(D\text{-}Ph\text{-}TRZ)$, where $E(D\bullet)$ is the ground-state energy of the radical fragment of the donor after C–N bond homolysis, $E(\bullet Ph\text{-}TRZ)$ is the ground-state energy of the radical fragment of Ph-TRZ after C–N bond homolysis, and $E(D\text{-}Ph\text{-}TRZ)$ is the ground-state energy of the intact TADF molecule before homolysis. $\Delta H_{C-N}$ was estimated from BDE and the $^1ICT$ transition state energy for each conformer.

**Photolysis**. Ar-saturated THF solutions containing 100 μM of each compound were added to a quartz cell (path length = 1.0 cm) and photoirradiated with broadband light by using a Xe lamp (300 W, Asahi Spectra, Max 303) placed 6 cm from the cell. The UV–vis absorption and photoluminescence spectra were recorded during photolysis. Photolysis was repeated with an Ar-saturated THF solution (500 μM in a 10 mL vial, o.d. = 1.7 cm) under the same condition. An aliquot of

100 μL was taken which was diluted in 400 μL $CH_3CN$ (HPLC grade, JT Baker). A volume of 5 μL was injected into a HPLC–ESI-mass spectrometer (Agilent, 1260 Infinity II/6120 Quadrupole LC/MS; VWD at 254 nm). The change in the dopant concentration was monitored by using the benzophenone internal standard (200 μM) which was delivered after the photolysis. A mixture of $H_2O$ and $CH_3CN$ was employed as the eluent.

**Determination of quantum yields of intrinsic degradation**. The quantum yields for intrinsic degradation ($QY$) of the dopants were determined by the standard ferrioxalate actinometry. A 0.0060 M $K_3[Fe(C_2O_4)_3]$ solution served as the chemical actinometer. Five hundred microliters of the $K_3[Fe(C_2O_4)_3]$ solution was transferred to a 1 cm × 1 mm quartz cell, and the solution was photoirradiated with broad-band light from an Xe lamp for 20 s. Then same amount of 1% 1,10-phenanthroline in sodium acetate buffer solution (4.09 g $CH_3COONa$ dissolved in 18 mL of 0.5 M $H_2SO_4$ and 32 mL of milli-Q water) were added and stored under dark for 1 h. The absorbance change at 510 nm was recorded. Inserting the value to Eq. (1) returned the light intensity value of $1.9 \times 10^{-7}$ einstein $s^{-1}$:

$$\text{Light intensity}\left(I_0,\ \text{einstein s}^{-1}\right) = (\Delta Abs(510\,\text{nm}) \times V)/\left(QY \times 11050\,\text{M}^{-1}\,\text{cm}^{-1} \times \Delta t\right). \tag{1}$$

In Eq. (1), $\Delta Abs(510\,\text{nm})$, $V$, $QY$, and $\Delta t$ are the absorbance change at 510 nm, volume (L), the quantum yield (1.1) of the ferrioxalate actinometer at 420 nm[43], and photoirradiation time (s), respectively. A 2.0 mL THF solution containing 100 μM dopant was photoirradiated under the identical condition. The decrease in the dopant concentration was quantitated by the HPLC technique described above, and inserted into Eq. (2):

$$QY = \left\{ ([\text{dopant}]_0 - [\text{dopant}]) \times V \right\} / (I_0 \times \Delta t) \tag{2}$$

In Eq. (2), $[\text{dopant}]_0$ and $[\text{dopant}]$ are the molar concentrations of the dopant before and after the photolysis, $\Delta t$ (s) is the photoirradiated time, $V$ is the volume of the solution (L), and $I_0$ are the light intensity obtained by Eq. (1) (einstein $s^{-1}$).

**Variable-temperature $^1H$ NMR experiments**. Variable-temperature $^1H$ NMR (300 MHz) experiments were performed with a Bruker, AVANCE III 300 spectrometer. 2.0 mM AZP-TRZ was dissolved in DMSO-$d_6$. $^1H$ NMR spectra were obtained at 373, 353, 333, 313, and 289 K.

**Electroluminescence**. The OLEDs were fabricated on indium tin oxide (ITO) substrates precleaned by performing ultrasonic treatments in acetone, iso-propanol, and deionized water. Each cleaned ITO substrate was treated with oxygen plasma prior to the deposition of the organic materials. The device structure ITO/DNTPD (60 nm)/BPBPA (20 nm)/PCzAC (10 nm)/mCBP:dopant (30 nm)/DBF-Trz (5 nm)/ZADN (30 nm)/LiF:Al was implemented in all devices. The organic materials, LiF, and Al were deposited at a base pressure of $1.0 \times 10^{-7}$ torr. All fabricated devices were protected from oxygen and moisture by encapsulation with a glass cover with a CaO getter attached. An epoxy adhesive was used to seal the glass cover and substrate. Device characterization was performed with a voltage sweep at an interval of 0.5 V. The electrical input source was a Keithley 2400 source measurement unit and the optical measurement system was a CS 2000 spectroradiometer. Device lifetimes were determined by using a lifetime measurement system equipped with a Si photodiode and a constant current input unit.

## Data availability

Supplementary Information includes Supplementary Figs. 1–33, displaying steady-state electronic spectra, voltammograms, fluorescence decay traces, photolytic degradation of films, intrinsic degradation of Cbz-TRZ, DMAC-TRZ, PXZ-TRZ, and PTZ-TRZ, operation lifetime and its concentration dependence of devices, correlations of stability and molecular factors, conformeric change of AZP-TRZ, ring strain, $s$-character in pseudo-equatorial forms, simulated electronic spectra, solvatochromism, a plausible mechanism for degradation of Cbz-TRZ, and $^1H$ and $^{13}C\{^1H\}$ NMR spectra; Supplementary Tables 1–6, listing photophysical data, calculated geometric and electronic parameters of the pseudo-axial and pseudo-equatorial forms, electroluminescence performances, crystallographic data for AZP-TRZ, and selected bond distances and bond angles for AZP-TRZ. The X-ray crystallographic coordinates for structures reported in this study have been deposited at the Cambridge Crystallographic Data Centre (CCDC), under deposition numbers CCDC-1951347. The cif file for AZP-TRZ is available as the Supplementary Data 1 or can be obtained free of charge from The Cambridge Crystallographic Data Centre via www.ccdc.cam.ac.uk/data_request/cif. Any further relevant data are available from the authors upon reasonable request.

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

## Acknowledgements

This work was supported by the Midcareer Research Program (NRF-2019R1A2C2003969) through National Research Foundation grants funded by the Ministry of Science, Information, and Communication Technology (ICT) and Future Planning (MSIP).

## Author contributions

S.H. designed and performed the most of the experiments, analyzed the data, and wrote the paper. Y.K.M. performed the electrochemical characterization. H.J.J. fabricated, tested and analyzed the devices. S.K. assisted with photolysis and analyzed the degradation byproducts. H.J. provided helpful comments for organizing the manuscript. J.Y.L. supervised the work at Sungkyunkwan University. Y.Y. coordinated all of the experiments and analyses. Y.Y. and J.Y.L. co-wrote the paper. All authors contributed to discussions regarding the study, and edited the paper.

## Competing interests

The authors declare no competing interests.
