## [Peer Review File · Communications Chemistry]

This manuscript has been previously reviewed at another Nature Research journal. This document only contains reviewer comments and rebuttal letters for versions considered at Communications Chemistry.

Reviewers' comments:

Reviewer #1 (Remarks to the Author):

I have read the previous referees' reports and the authors' responses. My impression is that the authors make a pretty good revision according to the referees' comments. In particular, the authors now acknowledge that the contribution of long-lived triplet excitons to device degradation has been considered and is important information to the general readers. The theoretical calculation results are reasonable. Therefore, in my opinion, the manuscript is now suitable for publication.

Reviewer #2 (Remarks to the Author):

I am able to the response from the authors; here are my comments for responses.

To the response to Comment 1:

I'd like to reiterate that the "common belief" of OLED material stability correlates the photo-stability of an amine-based emitter with the energy of its lowest singlet excited state, not the "emissive exciton" energy calculated from the EL peak wavelength. The EL spectrum of a TADF device are significantly affected by a number of factors, including ICT and conformation, and cannot be properly correlated with the S1 of emitters. Using energies from the EL spectrum to substitute S1 in the discussion of TADF OLED stabilities would be incorrect. The author already indicates that these emitters adopt to their favored conformation or conformation mixture in solution. The absorption edge in solution seems a better figure of merit to assess the S1. Even if there are some minor bands not well resolved close to the edge, the onset parts of these spectra are well separated (with the exception of DMAC-TRZ and PTZ-TRZ) to determine their S1 order.

To the response to Comment 2:

Simply put, molecular conformation can alter the S1 energy of these emitters, and the S1 of their favored conformer or conformer mixture determines their device lifetime. Neither of these findings is strictly new. However, this work still bear some significance by connecting the dots between these two facts. The potential design strategy implied in this study would be using conformation to lower the excited state of TADF emitters, thus increasing its lifetime, which seems pretty obvious and not particularly useful.

The responses of Comment 3 to 8 are fine. I appreciate the author's efforts.

But after re-read the paper, I also find these issues should be answered as follows:

-Regarding Figure 2e, (1) a purple line of Cbz-TRZ appears, but this compound has not been discussed throughout the text. (2) The vertical axis of the figure indicates that the photolysis experiment was performed at an initial concentration of 100 μM , but the text wrote, "Photolysis. Ar-saturated THF solutions containing 50 μM of each compound were added to a quartz cell (path length = 1.0 cm) and photoirradiated with broad-band light by using a Xenon lamp ", so at what concentration did the photolysis experiment begin? This involves the calculation of k_d . (3) The decay behavior of materials in solution and film may be very different, and the results of photo-aging experiments of the film will be more convincing. (4) The fitting results in the figure appear to be straight lines, which means that the photolysis kinetics of these materials are zero-order reactions. The k_d (lines 132-134) calculated in the paper also has the dimension of the zero-order reaction rate constant. (M / s). However, the reaction should be a first-order reaction, and k_d is also mentioned in the article as "The apparent rate of the unimolecular photolytic degradation", suggesting that this is a single molecule photolysis; please explain why the experimental result is an apparent zero-order reaction.

-Regarding Figure 3e, (1) the legend is "Electroluminescence spectra of devices", but the vertical axis is "Photoluminescence intensity". Is the figure an EL spectrum or a PL spectrum? (2) In this figure, four compounds The positions of the peaks of the γ and λ_{em} of the corresponding solutions in Table 1 are greatly different (the general deviation is more than 50 nm), and some of the luminescent red shifts, and some of the blue shifts. In addition, is λ_{em} in Table 1 measured in THF or toluene?

-In Table 2, the data of PXZ-TRZ and PTZ-TRZ are incorrect. For example, the " ΔH_{C-N} " of the PTZ-TRZ in the table under the pseudo axial form is -0.64 eV. However, this value is actually obtained by the difference between BDE and 1ICT (2.57 eV – 3.21 eV) of PXZ-TRZ under the structure, rather than the corresponding data of PTZ-TRZ (2.74 eV – 3.24 eV). Authors must carefully check the data.

Reviewer #3 (Remarks to the Author):

The authors revised the manuscript according to my review comments except for some. After addressing the followings, I recommend the publication of this work.

1) On my first comment, I agree with the revision of the title of this study. The inclusion of "Bearing Cycloamino Donors" is good, but more important point is that the donor has two conformers. Also, the term, "Molecular Factors" look vague. The factor clarified in this study is conformation. So, I recommend the title such as, "Conformation Dependent Degradation of Thermally Activated Delayed Fluorescence Materials Bearing Cycloamino Donors."

2) On my second comment, I think that the authors' excuse, "there exists no available experimental method that can directly probe the conformation of dopants in thick amorphous films," looks reasonable. As far as I know, no one has experimentally determined conformations in any amorphous films. The assignments of 1H NMR spectra are not still clear for me, but the authors added additional discussion on the conformers.

3) On my third comment, p. 8 line 178: There are no experimental data on dopant concentrations.

I cannot agree with the authors' opinion. Experiments on doping concentrations are inevitable to describe, "This observation supports the claim that for device stability the reactivity of the individual excitonic state is more important than the dopant concentration."

4) One my forth comment, p. 9 line 205: Why is increased s-character of nitrogen the origin of higher stability?

Many factors influence the values of $1/k_d$, LT97, etc. To clarify the relation between the s-character and stability, the authors should show the data on the stability or reactivity of the materials (not the devices) with different s-characters.

Others look fine.

We appreciate the reviewers for sharing their valuable time with our manuscript. We made the following changes in response to the referees' comments: The comments of the referees are in italics, and our responses are in blue.

Reviewer 1

Comments:

I have read the previous referees' reports and the authors' responses. My impression is that the authors make a pretty good revision according to the referees' comments. In particular, the authors now acknowledge that the contribution of long-lived triplet excitons to device degradation has been considered and is important information to the general readers. The theoretical calculation results are reasonable. Therefore, in my opinion, the manuscript is now suitable for publication.

Response: We appreciate the reviewer for the positive comments. Since this reviewer does not request additional revisions, we revised our manuscript based on the comments of the reviewers 2 and 3.

Reviewer 2

Comments:

I'd like to reiterate that the "common belief" of OLED material stability correlates the photo-stability of an amine-based emitter with the energy of its lowest singlet excited state, not the "emissive exciton" energy calculated from the EL peak wavelength. The EL spectrum of a TADF device are significantly affected by a number of factors, including ICT and conformation, and cannot be properly correlated with the S₁ of emitters. Using energies from the EL spectrum to substitute S₁ in the discussion of TADF OLED stabilities would be incorrect. The author already indicates that these emitters adopt to their favored conformation or conformation mixture in solution. The absorption edge in solution seems a better figure of merit to assess the S₁. Even if there are some minor bands not well resolved close to the edge, the onset parts of these spectra are well separated (with the exception of DMAC-TRZ and PTZ-TRZ) to determine their S₁ order.

Response and revisions: We thank the reviewer for sharing his or her valuable time with our manuscript. Following this comment, we have newly determined the singlet excited state (S₁) energy using the UV-vis absorption spectra of the dopant solutions. A variety of organic solvents were employed for the measurements, because some dopants displayed poor spectral resolutions in certain solvents. We learned that ethyl acetate is the best solvent that enabled location of the onset wavelength. The S₁ energy was calculated with the absorption onset wavelength. The five dopants possessed the following S₁ energy ordering: Cbz-TRZ (3.16 eV) > 3.14 eV (PTZ-TRZ) > 3.12 eV (AZP-TRZ) > 2.76 eV (DMAC-TRZ) > 2.59 eV (PXZ-TRZ). As shown below, the S₁ energies display poor correlations with the intrinsic stability (i.e., 1/k_a) and the operational lifetime (LT₉₇).

Plots of the intrinsic stability ($1/k_d$, blue triangles) and the operational lifetime (LT_{97} , red circles) as a function of the singlet excited state (S_1) energies of the compounds.

The newly obtained UV-vis absorption spectra have been included in our revised Supplementary Fig. 1. The above plot has also been included in our revised Supplementary Fig. 11.

Supplementary Figure 1. Steady-state electronic spectra. (a) UV-vis absorption and (b) photoluminescence spectra of EtOAc solutions containing 50 μM compounds (298 K). Excitation wavelengths: Cbz-TRZ, 363 nm; AZP-TRZ, 372 nm; DMAC-TRZ, 379 nm; PXZ-TRZ, 414 nm; PTZ-TRZ, 359 nm.

Supplementary Figure 11. Correlating stability and molecular factors. (a) Plots of the intrinsic stability ($1/k_d$, blue triangles) and the operational lifetime (LT_{97} , red circles) as a function of the singlet excited state (S_1) energies of the compounds. (b) Plots of the intrinsic stability ($1/k_d$, blue triangles) and the operational lifetime (LT_{97} , red circles) as a function of the natural decay rate ($1/\tau_{\text{avg}}$) of the compounds. (c) Plots of the radiative (k_r) and non-radiative (k_{nr}) rate constants for prompt (black symbols) and delayed fluorescence (pink symbols) as a function of $1/k_d$. (d) Plots of the radiative (k_r) and non-radiative (k_{nr}) rate constants for prompt (black symbols) and delayed fluorescence (pink symbols) as a function of LT_{97} . (e) Plots of the intrinsic stability ($1/k_d$, blue triangles) and the operational lifetime (LT_{97} , red circles) as a

function of the quantum yield for rISC (Φ_{rISC}). (f) Plot of Φ_{rISC} as a function of the Boltzmann factor of the pseudo-equatorial form and the pseudo-axial form ($K_{\text{eq/ax}}$).

2. Simply put, molecular conformation can alter the S1 energy of these emitters, and the S1 of their favored conformer or conformer mixture determines their device lifetime. Neither of these findings is strictly new. However, this work still bears some significance by connecting the dots between these two facts. The potential design strategy implied in this study would be using conformation to lower the excited state of TADF emitters, thus increasing its lifetime, which seems pretty obvious and not particularly useful. The responses of Comment 3 to 8 are fine. I appreciate the author's efforts.

Response: We appreciate the reviewer for the positive comments. Because this comment does not include an additional revision request, we revised our manuscript based on the following comments.

3. Regarding Figure 2e, (1) a purple line of Cbz-TRZ appears, but this compound has not been discussed throughout the text. (2) The vertical axis of the figure indicates that the photolysis experiment was performed at an initial concentration of 100 μ M, but the text wrote, " Photolysis. Ar-saturated THF solutions containing 50 μ M of each compound were added to a quartz cell (path length = 1.0 cm) and photoirradiated with broad-band light by using a Xenon lamp ", so at what concentration did the photolysis experiment begin? This involves the calculation of k_d . (3) The decay behavior of materials in solution and film may be very different, and the results of photo-aging experiments of the film will be more convincing. (4) The fitting results in the figure appear to be straight lines, which means that the photolysis kinetics of these materials are zero-order reactions. The k_d (lines 132-134) calculated in the paper also has the dimension of the zero-order reaction rate constant. (M / s). However, the reaction should be a first-order reaction, and k_d is also mentioned in the article as "The apparent rate of the unimolecular photolytic degradation", suggesting that this is a single molecule photolysis; please explain why the experimental result is an apparent zero-order reaction.

Response to comment (1): The following results of Cbz-TRZ have included in our revised manuscript:

- 1) Photophysical data (Table 1)
- 2) Intrinsic stability of Cbz-TRZ (Fig. 2e)
- 3) Synthesis details (Methods) and spectroscopic identification data (Supplementary Figs. 18 and 19)
- 4) UV-vis absorption and photoluminescence spectra (Supplementary Fig. 1)
- 5) Cyclic and difference pulse voltammetry (Supplementary Fig. 2)
- 6) Fluorescence lifetimes (Supplementary Fig. 3)
- 7) Photolysis behaviors (Supplementary Fig. 5)
- 8) Simulated electronic energies (Supplementary Table 1)
- 9) Simulated geometry parameters and the bond dissociation energy (Supplementary Table 2)
- 10) Proposed mechanism for degradation of Cbz-TRZ (Supplementary Fig. 17)

In addition, Figure 1 has been revised to include the chemical structure of Cbz-TRZ.

Response to comment (2): We have corrected the error. 100 μM solutions were employed throughout the photolysis experiments.

Response to comment (3): Following the suggestion, we prepared thick mCBP films doped with 10 wt % dopants on glass substrates through vacuum evaporation. The films were photoirradiated under illumination with broad-band light from a Xe lamp (300 W). The progress of the photolysis was monitored using UV-vis absorption spectroscopy. We found that photolysis of the films was very sluggish relative to the case of the solutions. This weak reactivity retarded quantification of the degradation process using HPLC techniques. In the UV-vis absorption spectra, a hypochromic shift in the intramolecular charge-transfer (ICT) transition band of AZP-TRZ was observed during photolysis. On the contrary, the other dopant films did not show noticeable decreases in their ICT transition bands because they are offset by the more intense increases of the high-energy bands that appear in the UV regions. Although the identical increase in the high-energy band was found in the photolysis of the dopant solutions (Fig. 2a), it cannot serve as a measure of the dopant degradation in films because the spectral signatures of the mCBP host decomposition occur in the same region. Therefore, we would like to keep the solutions data in our revised manuscript because they enable more reliable quantification of the extent of degradation.

Supplementary Figure 4. Photolytic degradation of films. UV-vis absorption difference spectra of thick mCBP films containing 10 wt % dopants recorded under photoillumination of broad-band light from a Xe lamp (300 W): (a) AZP-TRZ, (b) Cbz-TRZ, (c) DMAC-TRZ, (d) PTZ-TRZ, and (e) PXZ-TRZ.

The film results have been included in Supplementary Fig. 4. We also added the following discussion in our revised manuscript:

“An identical trend was also found for vacuum evaporated thick films of mCBP doped with 10 wt % AZP-TRZ (Supplementary Fig. 4).”

Response to comment (4): This comment is correct. The photolytic decomposition should follow the first-order kinetics, as the reviewer indicated. However, the kinetics data shown in Fig. 2e appear in the zero order. In order to explain this discrepancy, we considered mass balances for a ground-state (A) and an excited-state dopants (A*) based on the following equation:

In this equation, k_A is the rate constant for photoexcitation of A, k_A^{-1} , is the rate constant for natural decay of A*, and k_D is the rate constant for unimolecular decomposition from A*. The time-dependent changes in the concentrations of A and A* are expressed as eqs 1 and 2:

$$\frac{d[A]}{dt} = -k_A[A] + k_A^{-1}[A^*] \text{ (eq 1)}$$

$$\frac{d[A^*]}{dt} = k_A[A] - k_A^{-1}[A^*] - k_D[A^*] \text{ (eq 2)}$$

Eqs 1 and 2 are serial ordinary differential equations of the first order. Therefore, they are analytically soluble. The solution for [A] is

$$[A] = \left[\frac{k_A + \beta}{\beta - \alpha} \cdot \exp(\alpha t) - \frac{k_A + \alpha}{\beta - \alpha} \cdot \exp(\beta t) \right] [A]_0$$

, where α and β are

$$\alpha = \frac{-(k_A + k_A^{-1} + k_D) + \sqrt{(k_A + k_A^{-1} + k_D)^2 - 4k_A k_D}}{2}$$

$$\beta = \frac{-(k_A + k_A^{-1} + k_D) - \sqrt{(k_A + k_A^{-1} + k_D)^2 - 4k_A k_D}}{2}$$

, and $[A]_0$ is the initial concentration of A.

Considering $k_A \gg k_A^{-1} \gg k_D$, we could simplify the solution [A] into

$$[A] = \left[\frac{k_A^{-1} + k_D}{2k_A} \cdot \exp\left(-\frac{k_A^{-1} + k_D}{2} t\right) + \frac{2k_A - k_A^{-1} - k_D}{2k_A} \cdot \exp\left(-\frac{2k_A + k_A^{-1} + k_D}{2} t\right) \right] [A]_0$$

One can expect from the solution of [A] that the decrease in the dopant decomposition will follow the biexponential decay kinetics. The second exponential term corresponds to the

formation of the dopant exciton, and the first exponential term corresponds to the decrease in $[A]_0$ through the dopant exciton. Note that the latter (i.e., the first exponential term) is mainly governed by the natural decay with the rate constant k_A^{-1} . Therefore, it is probable that the intrinsic degradation with the rate constant k_D may not appear as the first-order decay kinetics during the overall photolytic decomposition. Please also note that the HPLC quantification is not so accurate to track the curvature of the decay curve, although the AZP-TRZ traces exhibit very small deviations from linearity. This limitation forced us to make linear fits at the initial data points. However, this linear fit does not necessarily imply a zero-order kinetics. In order to clarify this point, we have added the following sentence in our revised manuscript:

“The apparent rate of the unimolecular photolytic degradation (k_d) was determined from the linear fit of the initial points of the AZP-TRZ concentration. Note that this linear fit does not necessarily indicate a zero-order kinetics.”

Finally, we thank the reviewer again for his or her insightful comments.

4. Regarding Figure 3e, (1) the legend is “Electroluminescence spectra of devices”, but the vertical axis is “Photoluminescence intensity”. Is the figure an EL spectrum or a PL spectrum? (2) In this figure, four compounds The positions of the peaks of the γ and λ_{em} of the corresponding solutions in Table 1 are greatly different (the general deviation is more than 50 nm), and some of the luminescent red shifts, and some of the blue shifts. In addition, is λ_{em} in Table 1 measured in THF or toluene?

Response and revisions: The error in the legend in Fig. 3e has been corrected. They are the electroluminescence spectra. As the reviewer pointed out, some of the electroluminescence spectra appear in the regions different from those of solution photoluminescence spectra. This difference may be due to conformeric heterogeneity which depends sensitively on the rigidity and polarity of medium. For example, PTZ-TRZ displays fluorescence emission in the near-UV region in EtOAc solution with a peak wavelength of 435 nm, whereas it produces orange electroluminescence emission with a peak wavelength of 547 nm. The orange emission is actually observed as a shoulder in the fluorescence spectrum (Supplementary Fig. 1), suggesting that the conformeric heterogeneity is responsible for the differed emission behaviors. Finally, Table 1 have been updated to contain the peak emission wavelengths obtained in EtOAc.

5. In Table 2, the data of PXZ-TRZ and PTZ-TRZ are incorrect. For example, the “ $\Delta HC-N$ ” of the PTZ-TRZ in the table under the pseudo axial form is -0.64 eV. However, this value is actually obtained by the difference between BDE and ICT (2.57 eV – 3.21 eV) of PXZ-TRZ under the structure, rather than the corresponding data of PTZ-TRZ (2.74 eV – 3.24 eV). Authors must carefully check the data.

Response and revisions: We deeply appreciate the reviewer for careful reading. We have thoroughly examined the values, and revised the incorrect calculations.

Reviewer 3

Comments:

1. On my first comment, I agree with the revision of the title of this study. The inclusion of “ Bearing Cycloamino Donors ” is good, but more important point is that the donor has two conformers. Also, the term, “ Molecular Factors ” look vague. The factor clarified in this study is conformation. So, I recommend the title such as,
“ Conformation Dependent Degradation of Thermally Activated Delayed Fluorescence Materials Bearing Cycloamino Donors. ”

Response and revisions: We appreciate the reviewer for the suggestion. Following the suggestion, we have changed the title of our revised manuscript.

2. On my second comment, I think that the authors’ excuse, “ there exists no available experimental method that can directly probe the conformation of dopants in thick amorphous films, ” looks reasonable. As far as I know, no one has experimentally determined conformations in any amorphous films. The assignments of ^1H NMR spectra are not still clear for me, but the authors added additional discussion on the conformers.

Response: As the reviewer commented, we have included more discussion for the ^1H NMR results in our first revision. Since the reviewer did not request additional experiments, we would like to keep the original discussion.

3. On my third comment, p. 8 line 178: There are no experimental data on dopant concentrations. I cannot agree with the authors’ opinion. Experiments on doping concentrations are inevitable to describe, “ This observation supports the claim that for device stability the reactivity of the individual excitonic state is more important than the dopant concentration. ”

Response and revisions: We thank the reviewer for the valuable suggestion. Based on the comment, we have performed device experiments with varying the doping concentration of TADF emitters (5, 10 and 20 wt %). Our original manuscript involved the device results that were obtained at a 10 wt % doping concentration only. The newly performed experiments revealed that operational lifetime of devices indeed increased in proportion with dopant concentrations. Decay profiles of relative luminance as a function of operation time are shown below (see Supplementary Fig. 10 attached in the next page). The concentration dependence indicates that i) transport along dopants or ii) recombination within dopants of charge carriers are intimately linked to operational stability. Note that all the dopants may facilitate electron carrier transport, judging from the large mismatch between the LUMO energies of the mCBP host and the DBF-Trz hole-blocking layer (Fig. 3a). Given that, increased dopant concentrations reduce the bulk density of positive polarons and the interfacial density of negative polarons. This balanced transport and facilitated injection of charge carriers mediated by dopants suppresses polaronic degradation. Therefore, it is plausible that degradation mechanisms should involve polaron, as well as the exciton of the pseudo-axial conformer, as destructive intermediates. We again thank the reviewer for the insightful comments.

Nevertheless, the concentration effect does not rebut our original finding that the operational lifetime correlates with the intrinsic stability of the dopant ($1/k_d$). The LT_{97} ordering remains unchanged at identical doping concentrations of the dopants (Supplementary Fig. 10a; see the next page). In addition, unstable dopants, such as AZP-TRZ, show a weak concentration-dependent increase in LT_{97} . As shown in Supplementary Fig. 10b, the relative increase of LT_{97} decreases as the intrinsic stability of the dopant decreases (see the next page). These two observations strongly indicate that the operational stability of devices is closely associated with the intrinsic stability of dopants.

The new results have been included in Supplementary Fig. 10. We also added the following sentence to highlight the effect of the doping concentration:

“The LT_{97} values increase with the doping concentration of the dopant. However, the concentration effect becomes insignificant as the intrinsic stability of the dopant decreases (Supplementary Fig. 10). This observation indicates the decisive role of the individual excitonic state in the device stability. This rational is actually consistent with the weak adherence of the LT_{97} trend to the offsets between the HOMO energies of the dopant and host materials (i.e., the extent of trap-assisted recombination of the dopant).”

Supplementary Figure 10. Concentration dependence. (a) $L_{T_{97}}$ of the devices with emitting layers of 5, 10, and 20 wt % dopant concentrations. (b) Relative increases in $L_{T_{97}}$ of the devices relative to that of 5 wt % device. (c–f) Plots of luminance decays as functions of operation time (i), current densities as

functions of applied voltage (ii), and % luminance as functions of current density (iii) of the devices having 5, 10, and 20 wt % of dopants: c, AZP-TRZ; d, DMAC-TRZ; e, PTZ-TRZ; f, PXZ-TRZ. Changes in the % luminance were recorded during operation of the devices driven at following constant current densities defined at an initial luminance of 200 cd m^{-2} : AZP-TRZ, 0.033 mA cm^{-2} (5 wt %), 0.036 mA cm^{-2} (10 wt %), and 0.042 mA cm^{-2} (20 wt %); DMAC-TRZ, 0.040 mA cm^{-2} (5 wt %), 0.032 mA cm^{-2} (10 wt %), and 0.027 mA cm^{-2} (20 wt %); PXZ-TRZ, 0.017 mA cm^{-2} (5 wt %), 0.018 mA cm^{-2} (10 wt %), and 0.019 mA cm^{-2} (20 wt %); PTZ-TRZ, 0.022 mA cm^{-2} (5 wt %), 0.018 mA cm^{-2} (10 wt %), and 0.018 mA cm^{-2} (20 wt %).

4. *One my forth comment, p. 9 line 205: Why is increased s-character of nitrogen the origin of higher stability? Many factors influence the values of $1/k_d$, LT_{97} , etc. To clarify the relation between the s-character and stability, the authors should show the data on the stability or reactivity of the materials (not the devices) with different s-characters.*

Response and revisions: We appreciate the reviewer for the helpful suggestion. Following the comment, we correlated the s-character with intrinsic stability ($1/k_d$) of the materials. We found positive linearity of the s-character with $1/k_d$ (Supplementary Fig. 14b; see below). This positive relationship is due to the fact that the increased s-character corresponds to the stronger C–N bond that is prone to excitonic cleavage. In addition, the pseudo-equatorial form becomes more stable at greater s-character. The latter effect directly linked to a greater population of the pseudo-equatorial form over the pseudo-axial form.

As the reviewer indicated, many factors influence the intrinsic stability of materials and the operational lifetime of devices. Our finding demonstrates that the s-character may serve as a useful measure of the intrinsic stability of the dopants. The new plot has been included in Supplementary Fig. 14.

Supplementary Figure 14. s-character in pseudo-equatorial forms. (a) Natural bond orbital analyses (CAM-B3LYP/6–311+G(d,p)) for the three C–N bonds in the pseudo-equatorial forms of the donor–TRZ compounds. The colored horizontal arrows indicate the increasing orders LT_{97} (violet), s-character (sky blue), the extent of effective conjugation in the donor (green), and the relative stability of the pseudo-equatorial form over the pseudo-axial form (blue). (b) Correlations of the % s-character with the intrinsic

stability ($1/k_d$, blue) and the operational lifetime (LT_{97} , red). The increased s-character strengthens the C–N bond that is prone to excitonic cleavage. In addition, the pseudo-equatorial form becomes more stable at greater s-character. The latter effect directly linked to a more population of the pseudo-equatorial form over the pseudo-axial form.

I believe that we have addressed all the reviewers' comments. I hope that our revised manuscript is now acceptable for publication.

Sincerely yours,

Youngmin You
Professor

REVIEWERS' COMMENTS:

Reviewer #2 (Remarks to the Author):

I think the authors have try their best to revise the MS, although it is not perfect, its quality is improved and many obvious errors are corrected. Thus, I decide to approve this revision this time.

Reviewer #3 (Remarks to the Author):

According to the reviewers' comments, the authors reasonably revised the manuscript. As a result, this study is much improved. I recommend the publication of this work as is.